# m^6^A Reader YTHDC1 Impairs Respiratory Syncytial Virus Infection by Downregulating Membrane CX3CR1 Expression

**DOI:** 10.3390/v16050778

**Published:** 2024-05-14

**Authors:** Lucas W. Picavet, Ellen C. N. van Vroonhoven, Rianne C. Scholman, Yesper T. H. Smits, Rupa Banerjee, Sjanna B. Besteman, Mattheus C. Viveen, Michiel M. van der Vlist, Marvin E. Tanenbaum, Robert J. Lebbink, Sebastiaan J. Vastert, Jorg van Loosdregt

**Affiliations:** 1Center for Translational Immunology, University Medical Center Utrecht, 3584 CX Utrecht, The Netherlands; l.w.picavet@umcutrecht.nl (L.W.P.); e.c.n.vanvroonhoven@umcutrecht.nl (E.C.N.v.V.); r.c.scholman@umcutrecht.nl (R.C.S.); m.viveen@umcutrecht.nl (M.C.V.); mvlist2@umcutrecht.nl (M.M.v.d.V.); b.vastert@umcutrecht.nl (S.J.V.); 2Hubrecht Institute-KNAW and University Medical Center Utrecht, 3584 CX Utrecht, The Netherlandsm.tanenbaum@hubrecht.eu (M.E.T.); 3Oncode Institute, 3584 CX Utrecht, The Netherlands; 4Department of Bionanoscience, Delft University of Technology, 2600 AA Delft, The Netherlands; 5Department of Medical Microbiology, University Medical Center Utrecht, 3584 CX Utrecht, The Netherlands; r.j.lebbink-2@umcutrecht.nl

**Keywords:** RSV, m^6^A, N6-methyladenosine, YTHDC1, CX3CR1

## Abstract

Respiratory syncytial virus (RSV) is the most prevalent cause of acute lower respiratory infection in young children. Currently, the first RSV vaccines are approved by the FDA. Recently, N6-methyladenosine (m^6^A) RNA methylation has been implicated in the regulation of the viral life cycle and replication of many viruses, including RSV. m^6^A methylation of RSV RNA has been demonstrated to promote replication and prevent anti-viral immune responses by the host. Whether m^6^A is also involved in viral entry and whether m^6^A can also affect RSV infection via different mechanisms than methylation of viral RNA is poorly understood. Here, we identify m^6^A reader YTH domain-containing protein 1 (YTHDC1) as a novel negative regulator of RSV infection. We demonstrate that YTHDC1 abrogates RSV infection by reducing the expression of RSV entry receptor CX3C motif chemokine receptor 1 (CX3CR1) on the cell surface of lung epithelial cells. Altogether, these data reveal a novel role for m^6^A methylation and YTHDC1 in the viral entry of RSV. These findings may contribute to the development of novel treatment options to control RSV infection.

## 1. Introduction

Respiratory syncytial virus (RSV) is the most prevalent cause of acute lower respiratory infection in young children and can also affect the elderly. An estimated 33 million RSV-associated acute lower respiratory infection episodes occur annually, contributing to more than 100,000 deaths globally [1]. Recently, there have been promising developments for RSV vaccines, with the US FDA approving the adjuvanted vaccine Arexvy and bivalent recombinant protein subunit vaccine ABRYSVO for vaccination in older adults and ABRYSVO now being registered for maternal immunization in Europe [2,3,4]. A better understanding of the molecular pathways that regulate RSV infection and disease pathogenesis is necessary to fill knowledge gaps and improve treatment.

RSV viral entry is mediated by its glycoproteins F and G, which are located on the virion surface and which are both able to independently interact with cellular heparan sulfate proteoglycans (HSPGs) on the cell surface to initiate infection [5,6]. Furthermore, several proteins have been demonstrated to promote virus entry, including insulin-like growth factor-1 receptor (IGF1R), epidermal growth factor (EGFR), nucleolin and intercellular adhesion molecule-1 (ICAM-1, in cell lines) [7,8,9,10,11]. Also, CX3C chemokine receptor 1 (CX3CR1) has been demonstrated to be utilized as an entry receptor in well-differentiated primary bronchial epithelial cultures as the RSV G protein contains a chemokine-like motif that can bind CX3CR1 [11,12]. 

In recent years different studies have identified a role for N6-methyladenosine (m^6^A) RNA methylation in the regulation of the viral life cycle of different viruses, including RSV [12,13,14,15,16,17,18,19,20,21,22,23,24]. Multiple m^6^A sites have been identified in various genes within the RSV genome (within the NS2, N, P, M, G and L genes) and anti-genome (N, P, G and F) [21]. These RSV m^6^A modifications prevent the recognition of the viral RNA by cytosolic pattern recognition receptor (PRR) retinoic acid-inducible gene I (RIG-I) and thereby prevent the initiation of host interferon responses [25,26,27]. Moreover, RSV infection alters the m^6^A landscape on host cell RNA, a phenomenon also observed in other RNA viruses [13,21,28,29].

In addition, it was demonstrated that cytoplasmic m^6^A reader proteins YTH domain-containing family-1 (YTHDF1), -2 and -3 can associate with methylated RSV RNA and positively regulate viral replication. These m^6^A reader proteins are mostly associated with the decay of RNA transcripts. YTH domain-containing protein 1 (YTHDC1), the main nuclear m^6^A reader protein with roles in splicing and nuclear export of RNA transcripts, has not been associated with anti-viral immunity and RSV so far [30,31]. This is where a gap in important knowledge was observed.

Therefore, we studied YTHDC1 regulatory function in the context of RSV infection. We demonstrate that YTHDC1 manipulation in lung epithelial cells directly affects RSV viral load. More specifically, we show that YTHDC1 suppresses the early infection of RSV via downregulation of CX3CR1 on the host cell. Altogether, these data demonstrate a novel role for YTHDC1 in the viral entry of RSV. A better understanding of the m^6^A-mediated regulation of RSV infection can contribute to the development of novel therapeutic strategies for the treatment of RSV-infected individuals.

## 2. Materials and Methods

### 2.1. Cell Culture

Human A549 cells and Vero cells were cultured in Dulbecco’s Modified Eagle Medium with GlutaMAX (DMEM, Fisher Scientific, Waltham, MA, USA, 12077549) supplemented with 10% heat-inactivated fetal bovine serum (FBS, Sigma-Aldrich, St. Louis, MO, USA) and 1% penicillin-streptomycin (P/S, Merck Life Science, Darmstadt, Germany, P0781-100ML) at 37 °C and 5% CO_2_. 

### 2.2. RSV-GFP Infection

A total of 100,000 A549 cells were cultured in wells of a 6-well plate and were washed with PBS (Sigma-Aldrich, St. Louis, MO, USA, D8537-24X500ML) prior to infection. Viral stocks of the A2 strain of RSV-expressing Green Fluorescent Protein (GFP) were thawed from −80 storage, and cells were infected with 1.0 MOI RSV-GFP in DMEM for 2 h at 37 °C and 5% CO_2_. 

After infection cells were washed with PBS three times before culturing in DMEM supplemented with 10% FBS 1% P/S for 72 h. Cells were detached with persistent resuspension before being transferred to 96-well plates for FACS staining. Specifically, cells were detached by pipetting up and down forcefully with a P1000 pipet. Subsequently, detachment was confirmed by visual inspection under a microscope to ensure the complete release of cells from the culture plate. GFP measurements were performed using a BD FACSCanto™ II flow cytometer, and FlowJo v10 was used for data analysis. The gating strategy for GFP-positive cells involved setting gates based on cells that were untransfected with RSV.

### 2.3. Manipulation of YTHDC1

For overexpression of YTHDC1, cells were cultured in DMEM without supplements. Cells were transfected with 2 ug DNA of either pcDNA3 empty vector, pcDNA3-YTHDC1 (Addgene, Watertown, MA, USA, 85167) overexpression vector, pLX304 empty vector or pLX304-YTHDC1 overexpression vector (Harvard University) using polyethylenimine (PEI MAX, Polysciences Europe, Baden-Wurttemberg, Germany, 24765-1). After 24 h, cells were washed with PBS, and the medium was refreshed with a supplemented DMEM culture medium. Cells were rested for 24 h before further procedures were performed. For knockdown of YTHDC1, cells were cultured in Opti-MEM reduced serum medium and transfected with either 100nM negative control siRNA (ON-TARGETplus non-targeting pool, Dharmacon, Lafayette, CO, USA, D-001810-10-05) or YTHDC1 siRNA (SMARTpool: siGENOME YTHDC1 siRNA, Dharmacon, M-015332-01-0005) using DharmaFECT 1 transfection reagent (Horizon Discovery LTD, Cambridge, UK, T-2001-02). Cells were washed with PBS, and the medium was refreshed with a supplemented DMEM culture medium after 24 h. Cells were rested for 24 h before further procedures were performed.

### 2.4. Analyzing CX3CR1 Expression Using Flow Cytometry

Cells were detached with persistent resuspension before being transferred to 96-well plates for FACS staining. Cells were fixed for surface staining using cytofix buffer (BD, 554655) for 30 min at 4 °C. Cells were stained with rat anti-CX3CR1 PE-Dazzle594 (monoclonal, Biolegend, Waltham, MA, USA, 341623) with 1% mouse serum (Bioconnect, Toronto, ON, USA, 88-NM35) for 30 min at 4 °C. Measurements were performed using a BD FACSCanto™ II flow cytometer, and FlowJo v10 was used for data analysis. 

### 2.5. Western Blot

For Western blot, cells were lysed in Laemmli buffer (0.12M Tris-Hcl, pH 6.8, 4% SDS, 20% Glycerol, 0.05 ug/uL bromophenol blue, 35 mM β-mercaptoethanol). Protein amounts were normalized using a BCA protein assay kit (Fischer, Waltham, MA, USA, 23225) and 20 ug protein per sample was used for SDS-PAGE separation. Blotting was performed utilizing a Polyvinylidene difluoride membrane (Immobilon-P PVDF 45 uM, Merck Chemicals, IPVH00010). The membranes were stained with indicated antibodies and enhanced chemiluminescence (ECL, Thermofisher, Waltham, MA, USA, 32106). Western blotting substrate was used to image blots using a Gel doc EZ imager (Biorad, South Granville, Australia). The following antibodies were used: rabbit anti-YTHDC1 (polyclonal, Abcam, Cambridge, UK, ab122340) and mouse anti-GAPDH (Life Technologies, Carlsbad, CA, USA, MA5-15738). 

### 2.6. Real-Time PCR

Cells were lysed in RLT buffer (RNeasy kit, Qiagen, Waltham, MA, USA, 74106), and RNA was isolated using the manufacturer’s protocol. cDNA synthesis was performed using an iScript cDNA synthesis Kit (Bio-Rad, South Granville, Australia). qPCR reaction was performed with SYBR Select master mix (Thermofisher, 13266519) and the following primer pairs: B2M-fw: TGCTGTCTCCATGTTTGATGTATCT; B2M-rev: TCTCTGCTCCCCACCTCTAAGT, YTHDC1-fw:; YTHDC1-rev:, RSV-NS1-fw: CAATTCATTGAGTATGATAAAAGTTAGATTACA, RSV-NS1-rev: AATATTATTATTAGGGCAAATATCACTACTTGTA; and CX3CR1-fw: AGTGTCACCGACATTTACCTCC, CX3CR1-rev: AAGGCGGTAGTGAATTTGCAC. 

### 2.7. Single Molecule Fluorescence In Situ Hybridization

Single-molecule fluorescence in situ hybridization is based on a protocol from Gasper et al. [32] smFISH probe generation Stellaris probe designer (https://www.biosearchtech.com/support/tools/design-software/stellarisprobe-designer accessed on 31 January 2022) was used to design probes that target the RSV genome. The probe sets consisting of 20 Mer oligonucleotides were ordered from Integrated DNA Technologies (IDT) and pooled. Cells were cultured in 96-well glass bottom plates. Cells were infected with 1.0 MOI RSV-GFP A2 strain within DMEM for 2 h at 37 °C and 5% CO_2_. Cells were fixed with 4% PFA in PBS for 10 minutes at room temperature. Cells were permeabilized with 100% ethanol for 30 minutes at 4 degrees. Probe sets were labeled as per protocol from Gasper et al. [32]. Genome probesets were labeled with Atto-565. Cell staining was done by Pacific Blue™ Succinimidyl Ester (Thermo Fisher). Images were collected 2 h post-infection by a Nikon TI inverted microscope with a perfect focus system, a Yokagawa CSU-X1 spinning disc, a 60× 1.4 NA oil objective and an iXon Ultra 897 EM-CCD camera (Andor) using NIS Elements Software (Nikon, Tokyo, Japan). Approximately 18 Z-slices were acquired at 0.8 μm intervals that covered the entire cell with a 50 ms exposure time. Maximum-intensity projection images were used for image analysis. Image analysis was performed with ImageJ software (https://imagej.net/ij/). The number of cells per image was counted blindly by hand, and the number of FISH spots was assessed using the built-in ‘particle analysis’ macro with equal color thresholds. 

## 3. Results

### 3.1. YTHDC1 Manipulation Influences RSV Gene Expression and Viral Infection

Previous studies have demonstrated that m^6^A RNA methylation of RSV RNA promotes viral replication [21]. However, the extent to which m^6^A modifications on host RNA can impact infection remains inadequately elucidated. Additionally, the involvement of the m^6^A reader protein YTHDC1 in RSV infection has not been evaluated. The nuclear m^6^A reader YTHDC1 has the capacity to modulate host RNA independent of its localization with cytoplasmic RSV RNA. 

To determine whether YTHDC1 impacts RSV viral load, we overexpressed YTHDC1 in lung epithelial cell line A549 and infected these cells with RSV-GFP (Figure 1A,F,G). Overexpression of YTHDC1 resulted in a reduced number of cells infected with RSV-GFP as determined by flow cytometry 72 h post-infection (Figure 1A,B). Also, the GFP expression per infected cell was significantly reduced, as indicated by reduced MFI levels in GFP-positive cells (Figure 1C,D). Similarly, RSV viral RNA expression was reduced by YTHDC1 overexpression measured by mRNA levels of RSV gene NS1 (Figure 1E). YTHDC1 mRNA and protein expression were not impacted by RSV-GFP infection (Figure 1H,I). Altogether, these results demonstrate that YTHDC1 overexpression impairs RSV infection or replication.

To validate that YTHDC1 manipulation affects RSV viral load, a siRNA-mediated knockdown approach was used. Endogenous YTHDC1 knockdown was performed in A549 cells, RSV-GFP virus was administered, and RSV viral load was measured by flow cytometry and qPCR. Knockdown of YTHDC1 resulted in an increased amount of RSV-GFP-positive cells and RSV-NS1 mRNA expression (Figure 1F,G and Figure 2A–D). Altogether, these data indicate that YTHDC1 suppresses RSV infection or replication. 

### 3.2. YTHDC1 Is Not Involved in the Activation of the Type 1 IFN Pathway 

m^6^A methylation of RSV RNA has been demonstrated to prevent host interferon responses [25,26,27]. We, therefore, hypothesized that the anti-viral effects of YTHDC1 are the result of increased IFN signaling. To test this, Ruxolitinib, a well-defined inhibitor of IFN-induced JAK signaling, was utilized to assess the effects of YTHDC1 on RSV infection in the absence of IFN signaling [33,34]. In the presence of Ruxolitinib, overexpression of YTHDC1 still suppressed RSV replication as measured by the amount of RSV-NS1 gene mRNA present (Figure 3A), suggesting that YTHDC1 did not affect RSV replication via IFN type I. To investigate this further, YTHDC1 was overexpressed in Vero cells, a cell line that lacks IFNa and IFNb genes [35] and is, therefore, more susceptible to viral infection [36]. YTHDC1 overexpression in these cells still resulted in a decreased expression of RSV-NS1 mRNA transcripts (Figure 3B), thereby validating our data in different cells and demonstrating that the effect of YTHDC1 on RSV infection is not mediated via type 1 IFN responses.

### 3.3. YTHDC1 Regulates Viral Entry of RSV 

To determine whether YTHDC1-mediated differences in viral load are the result of decreased RSV infection of host cells, we performed single-molecule RNA fluorescence in situ hybridization (smFISH) with probes against genomic RSV RNA 2 h post-infection [32]. This technique allows for visualization of early RSV infection. We found a negative correlation between YTHDC1 expression and the number of RSV-infected cells (Figure 4A–D). Knockdown of YTHDC1 resulted in an increased number of smFISH spots per cell, while a reduced number of spots per cell was observed upon overexpression of YTHDC1 in the host cells (Figure 4E). These data demonstrate that YTHDC1 manipulation affects viral infection already 2 h after infection. Because the amount of genomic RNA transcripts is affected early in infection, these results indicate a role for YTHDC1 in the regulation of viral entry. 

### 3.4. YTHDC1 Regulates the Expression of the RSV Entry Receptor CX3CR1

As we observed that YTHDC1 reduced RSV viral load early after infection, we next investigated the mechanism behind this observation. We hypothesized that YTHDC1 might affect the expression of RSV entry receptors and, specifically, CX3CR1. Overexpression of YTHDC1 significantly decreased the percentage of cells expressing CX3CR1 on the cell surface as measured by flow cytometry on non-permeabilized cells, and opposite effects were seen for YTHDC1 knockdown (Figure 5A,B). Strikingly, manipulation of YTHDC1 did not result in significant CX3CR1 mRNA expression changes, illustrating that the YTHDC1-dependent changes in CX3CR1 surface expression are not the result of altered transcription or RNA degradation (Figure 5C). Altogether, these data indicate that YTHDC1 can modulate the expression of RSV entry receptor CX3CR1 on the surface of lung epithelial cells, thereby hampering RSV entry. 

## 4. Discussion

In recent years, m^6^A RNA methylation has been implicated in the viral life cycle, including viral replication and survival. Furthermore, m^6^A has also been demonstrated to play a role in host anti-viral immunity against respiratory viruses such as human metapneumovirus (hMPV), Influenza A virus (IAV) and SARS-CoV-2 [23,25,37,38,39]. m^6^A modifications were previously identified on the RSV genome, but exactly how m^6^A is orchestrating the immune response of the host cell against RSV infection is still unknown. Here, we identify YTHDC1 as a novel negative regulator of RSV viral infection by downregulating the expression of host cell RSV entry receptor CX3CR1. 

Our study demonstrates that the m^6^A reader protein YTHDC1 impairs RSV infection and host cell entry. Research by Xue et al. demonstrated a positive effect of m^6^A on RSV replication [21]. In their study, METTL3 overexpression increased RSV viral load, whereas ALKBH5 overexpression induced opposite effects. These effects were mediated by binding of m^6^A readers YTHDF1, -2 and -3 to the methylated G gene of RSV, thereby mediating immune evasion [21,27]. In contrast, our data demonstrate that a m^6^A reader can also negatively impact RSV viral load. This difference might be explained by the fact that the role of YTHDC1 is specific for RSV entry and might not affect viral RNA itself. Accordingly, YTHDC1 and RSV RNA are located in different cellular compartments, predominantly nuclear and cytoplasmic, respectively; therefore, direct interaction of YTHDC1 with RSV RNA is unlikely [30,31]. A recent study strengthens the belief that YTHDC1 remains localized in the nucleus during RSV infection. Here, A549 cells were inoculated with negative-sense single-stranded RNA virus Influenza (MOI 5). With confocal microscopy, YTHDC1 was visualized in the nucleus of these infected cells. However, more research is needed to understand exactly under which circumstances or at which timepoints specific m^6^A reader proteins fulfill a certain role in RSV infection or spread.

We demonstrated that YTHDC1 impairs early RSV infection prior to viral replication. With the use of smFISH, we were able to detect the anti-viral effects of YTHDC1 as early as 2 h after RSV infection, allowing us to study viral entry. CX3CR1 is a well-known entry receptor for RSV infection that interacts with the CX3C chemokine motif of the RSV G protein necessary for binding to the receptor [40,41]. In previous research, it was found that the level of RSV replication correlates with the percentage of CX3CR1 expressing cells as RSV G protein works as a ligand-agonist for epithelial CX3CR1 [42]. This receptor interaction has become the target of many RSV vaccines, and the CX3C motif is important for the induction of protective immune responses [43,44]. We demonstrated that increased YTHDC1 levels reduced CX3CR1 protein expression and RSV viral load. Therefore, YTHDC1 manipulation could be an interesting new approach to regulate CX3CR1 expression and to prevent or reduce RSV infection. 

We acknowledge that our study harbors certain limitations. Firstly, while we demonstrate that YTHDC1 influences the membrane expression of CX3CR1, the exact molecular mechanism underlying how YTHDC1 modulates CX3CR1 membrane trafficking remains incompletely understood and warrants further investigation in the future. Furthermore, our study suggested a causal relationship between reduced CX3CR1 expression and diminished RSV entry, and we cannot exclude that YTHDC1 can also affect other targets involved in RSV viral entry, such as heparin sulfate. Nonetheless, there is a possibility that these observed phenomena might occur independently of each other. Another limitation arises from the cellular model employed in our research. A549 cells inherently express heparan sulfate proteoglycans (HSPGs) on their surface, which are known to facilitate RSV infection in various cell lines. Our study does not fully exclude the potential for RSV entry through receptors other than CX3CR1, including HSPGs. Additional studies are needed to delineate the relative contributions of different receptors in RSV infection.

In this study, we demonstrate that YTHDC1 modulates the (surface) expression of RSV entry receptor CX3CR1 without affecting CX3CR1 mRNA expression. We hypothesize that YTHDC1 inhibits CX3CR1 protein expression or the shuttling of CX3CR1 towards the cell membrane. In this context, YTHDC1 could potentially modulate the splicing or nuclear transport of mRNA transcripts encoding RNA binding proteins that aid in CX3CR1 translation or mRNA encoding regulatory proteins that are involved in the modulation of CX3CR1 trafficking to the cell membrane. However, further research is needed to unravel the exact role of YTHDC1 in CX3CR1 membrane expression.

To our knowledge, this is the first study that shows a relationship between nuclear m^6^A reader YTHDC1 and viral entry of RSV. We demonstrate a role for YTHDC1 in mediating CX3CR1 expression, thus regulating RSV viral entry. However, more research is needed to unravel the precise mechanism behind this observation. Furthermore, translating these findings to RSV infection in vivo, particularly in mouse models or human subjects, warrants further investigation. Such studies would be necessary to confirm the potential of targeting m^6^A modification pathways as a novel therapeutic or preventive strategy against RSV infections in humans. 

## Figures and Tables

**Figure 1 viruses-16-00778-f001:**
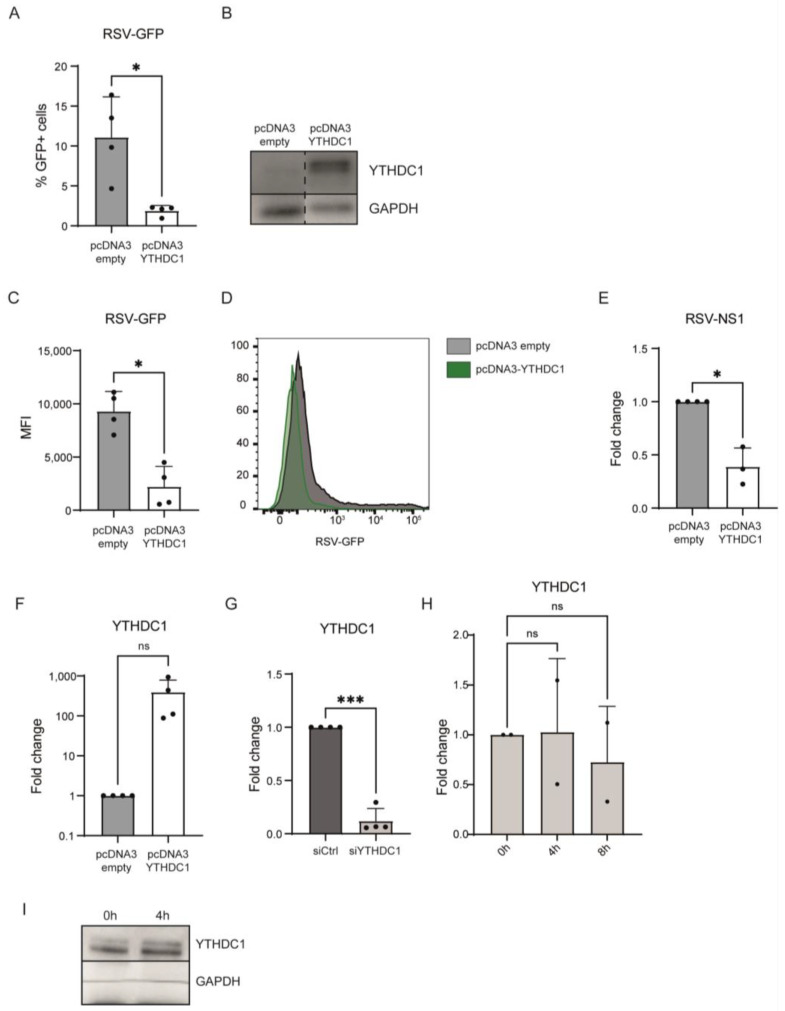
YTHDC1 overexpression downregulates RSV gene expression and viral load in A549 alveolar epithelial cells. (**A**) %RSV-GFP-positive cells 72 h post-RSV infection in empty vector control and YTHDC1 overexpression cells measured by flow cytometry (*p*-values were calculated using a paired *t*-test, * *p* < 0.05, n = 4). (**B**) Western blot analysis for YTHDC1 protein in empty vector control and YTHDC1 overexpression cells. (**C**,**D**) Mean fluorescent intensity (MFI) of RSV-GFP-positive cells (n = 4). (**E**) Relative mRNA expression of NS1-RSV gene 72 h post-RSV infection. Fold changes were calculated relative to housekeeping gene β2M. *p* values were calculated using a two-tailed unpaired *t*-test (* *p* < 0.001), n = 4. (**F**) Relative mRNA expression of YTHDC1 in empty vector control and YTHDC1 overexpression of A549 cells. Fold changes were calculated relative to housekeeping gene β2M. Fold changes were calculated relative to housekeeping gene β2M. *p*-values were calculated using a paired *t*-test, not significant (ns, *p* = 0.1418), n = 4. (**G**) Relative mRNA expression of YTHDC1 in negative control siRNA and YTHDC1 siRNA treated A549 cells measured with qPCR. *p*-values were calculated using a paired *t*-test, *** *p* < 0.005, n = 4. (**H**) A549 cells were infected with RSV-GFP. YTHDC1 expression was measured with qPCR at timepoints 0 h, 4 h and 8 h post-infection. *p*-values were calculated using one-way ANOVA with Geisser–Greenhouse correction and Dunnett’s multiple comparison test, ns *p* > 0.05, n = 2. (**I**) Western blot analysis of YTHDC1 expression in A549 cells infected with RSV-GFP 0 h and 4 h post-infection.

**Figure 2 viruses-16-00778-f002:**
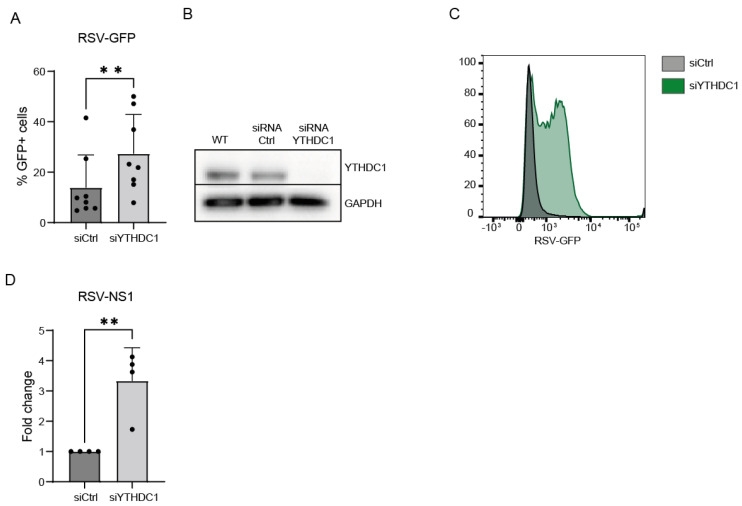
YTHDC1 knockdown upregulates RSV gene expression and viral infection. (**A**) Percentage RSV-GFP-positive cells 72 h post-RSV infection in YTHDC1 knockdown or control cells measured by flow cytometry (n = 8). (**B**) Representative Western blot analysis of YTHDC1 protein in YTHDC1 siRNA and negative control siRNA conditions. (**C**) Mean fluorescent intensity (MFI) of RSV-GFP-positive cells. (**D**) Relative mRNA expression of RSV-NS1 gene 72 h post-RSV infection measured with qPCR. Fold changes were calculated relative to housekeeping gene β2M. *p*-values were calculated using a two-tailed unpaired *t*-test; ** *p* < 0.01, n = 4.

**Figure 3 viruses-16-00778-f003:**
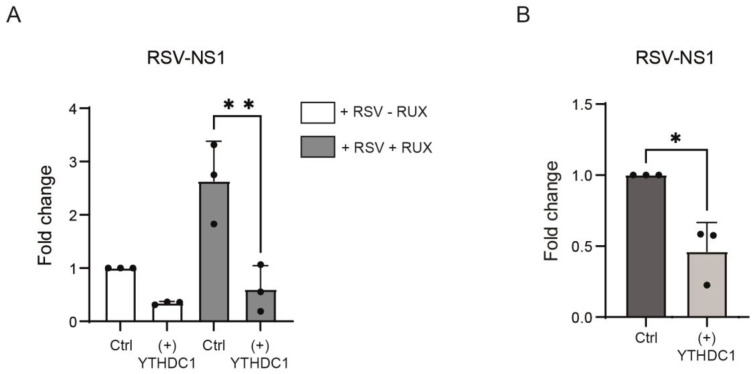
YTHDC1 acts independently of the type 1 interferon response. (**A**) A549 cells with overexpression of YTHDC1 or control vector were treated with 1 uM Ruxolitinib or DMSO and were infected with RSV-GFP. RSV-NS1 expression was measured with qPCR. *p*-values were calculated using a paired *t*-test, ** *p* < 0.01. (**B**) Vero cells YTHDC1 overexpression were treated with RSV-GFP for 72 h. RSV-NS1 expression was measured with qPCR. *p*-values were calculated using a paired *t*-test, * *p* < 0.05, n = 3.

**Figure 4 viruses-16-00778-f004:**
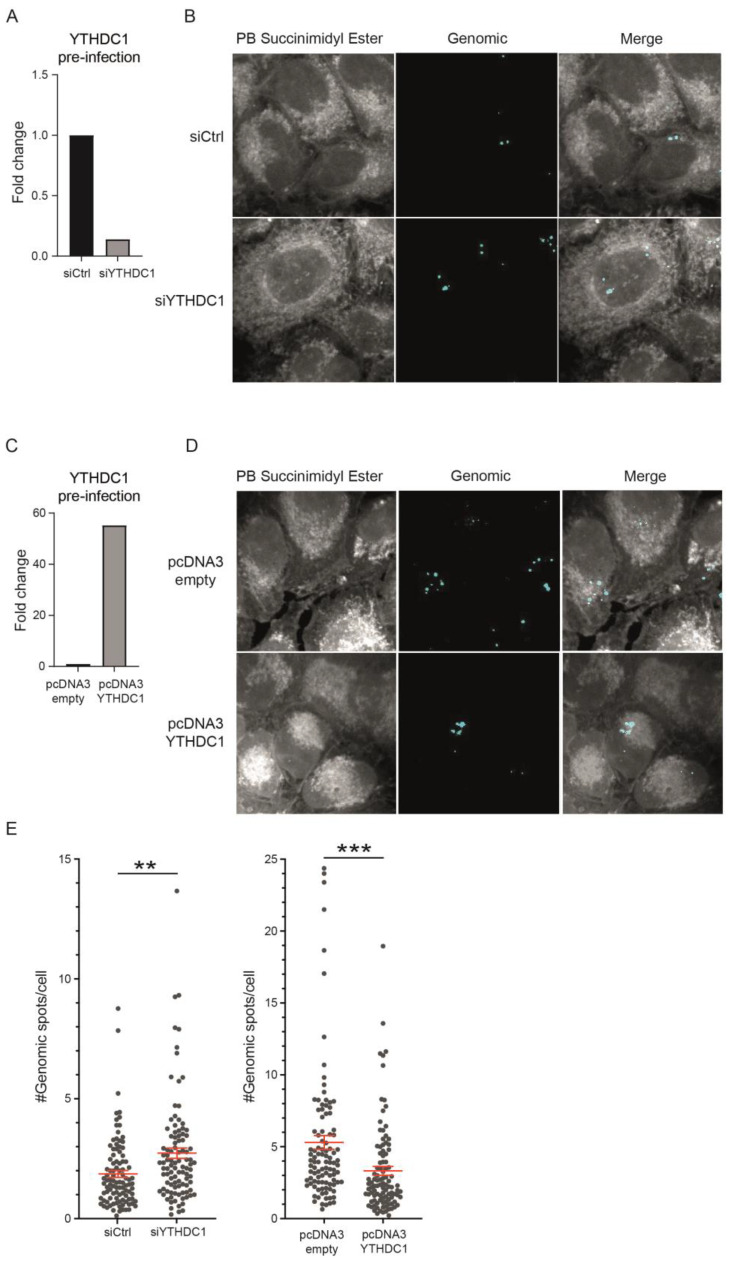
YTHDC1 regulates viral entry of RSV. smFISH analysis 2 h post-RSV infection of A549 cells with transient knockdown and overexpression of YTHDC1. (**A**,**C**) YTHDC1 mRNA expression in A549 cells with transient knockdown (**A**) or overexpression (**C**) of YTHDC1 prior to RSV infection. (**B**,**D**) Representative images showing Pacific Blue™ Succinimidyl Ester (gray), genomic-FISH (cyan) of smFISH analysis. (**E**) Genomic FISH spots per cell are shown for YTHDC1 overexpression and YTHDC1 knockdown conditions. *p*-values were calculated using a two-tailed two-sample *t*-test; ** *p* < 0.01, *** *p* < 0.001.

**Figure 5 viruses-16-00778-f005:**
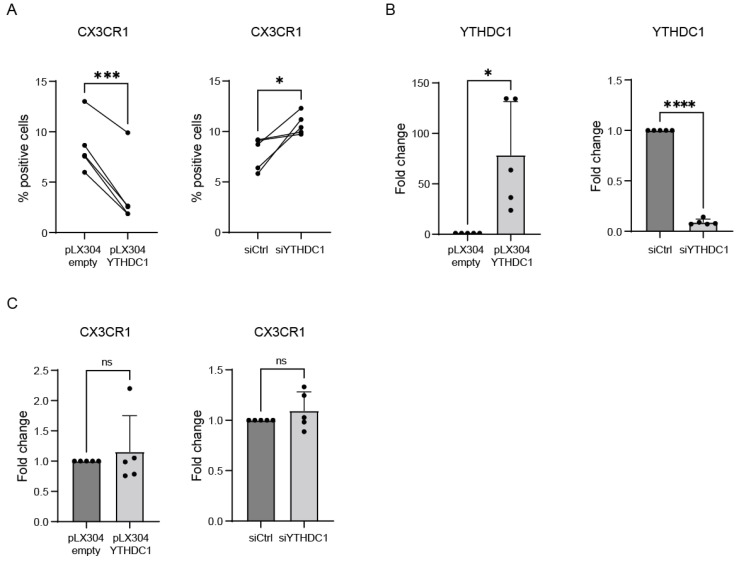
YTHDC1 regulates membrane expression of RSV entry receptor CX3CR1. (**A**) Percentage of surface CX3CR1 expressing A549 cells after YTHDC1 manipulation measured with flow cytometry (n = 4). (**B**) Transient overexpression or siRNA knockdown of YTHDC1 in A549 cells. YTHDC1 expression measured with qPCR (n = 5). (**C**) CX3CR1 expression in A549 cells with overexpression or knockdown of YTHDC1, measured with qPCR. Fold changes were calculated relative to housekeeping gene β2M. *p*-values were calculated using a two-tailed unpaired *t*-test; ns *p* > 0.05, * *p* < 0.05, *** *p* < 0.001, **** *p* < 0.0001, n = 5.

## Data Availability

All data generated or analyzed during this study are included in this article and Appendix A.

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
