# Peer review of "m6A Reader YTHDC1 Impairs Respiratory Syncytial Virus Infection by Downregulating Membrane CX3CR1 Expression"

_viruses, 2024, doi:10.3390/v16050778_

Round 1
Reviewer 1 Report
Comments and Suggestions for Authors

Reviewer 2 Report
Comments and Suggestions for Authors
Picavet et al show that m6a reader protein YTDHC1 has a negative impact on RSV entry and that this impact is mediated by the level of CX3CR1 and virus entry, and is independent of type I IFN and not associated with mRNA expression or stability. This is an interesting finding that will be of interest to the RSV field, as it improves understanding of the life cycle and provides potential drug or vaccine targets. Overall the study is clearly laid out. Up and down regulation of YTDHC1 is well controlled throughout, and there is a clear impact of the number of infected cells. The impact appears to be at the level of virus entry and to work via regulation of CX3CR1 levels. The quality of the work is high and the data are novel, but there are also some concerns.
* One concern with the major claim (YTDHC1 acts by downregulating CX3CR1 levels) is that the evidence is a bit thin, dependent on only one figure segment (Figure 4A). Is it possible for example, that YTDHC1 has an impact on heparin sulfate levels at the cell surface? Also the assay used is not specific to surface CX3CR1. Although the data do suggest it may work through CX3CR1 surface levels, it would require some more evidence to make a strong claim.
* In Fig. 1, can the authors exclude YTDHC1 induced cell toxicity as responsible for lower RSV-GFP levels? A cell viability assay would help to verify this. This is especially the case for overexpression. The lower level of GAPDH in the pcDNA3-YTHDC1 transfected cells in Fig. 1B could be an indication of reduced cell viability.
* To show CX3CR1 levels, the cells are "permeabilized for intracellular staining". This protocol does provide an indication of total cellular CX3CR1 levels but does not indicate surface levels. Ideally the staining should be done on non-permeabilized cells, as the cell surface levels determine the extent of RSV entry.
Minor comments
* In Fig 1A and C, GFP was counted every 4 hours for a 72 hour period. How was the GFP+ cell # derived? Which timepoint was used?
* on line 43, the data referenced (5,6) are from vero and hep2 cells, not lung epithelial cells.
* please describe RSV-GFP
* in general, more experimental details should be provided in the main text, rather than the reader piecing together the steps from the figure legends and materials and methods.
* please describe in more detail, how cells are processed for in situ data, following the 2 hour virus incubation.
* provide details for the process of cell detachment with persistent resuspension.
Round 2
Reviewer 1 Report
Comments and Suggestions for Authors The authors have done an impressive and comprehensive job of addressing most of my original concerns.Reviewer 2 Report
Comments and Suggestions for Authors
The revised manuscript by Picavet et al has improved, and most concerns are adequately addressed. The case for cytotoxity caused by YTDH-C1 (figure 1) remains weak; the added assay (figure A4) does help but does not necessarily distinguish metabolically active cells from non or less-metabolically active cells. The legend of new Figure 1I (western) incorrectly states that this was quantitated with qPCR.